# How do the prevalence and relative risk of non-suicidal self-injury and suicidal thoughts vary across the population distribution of common mental distress (the p factor)? Observational analyses replicated in two independent UK cohorts of young people

Ela Polek [1,2] Sharon A S Neufeld,[1] Paul Wilkinson,[1] Ian Goodyer,[1] Michelle St Clair,[3] Gita Prabhu,[4,5] Ray Dolan,[4,5] Edward T Bullmore,[1] Peter Fonagy,[6] Jan Stochl,[1,7] Peter B Jones[1,7]

For numbered affiliations see end of article.

**Correspondence to**
Professor Peter B Jones;
pbj21@cam.ac.uk

## ABSTRACT

**Objectives** To inform suicide prevention policies and responses to youths at risk by investigating whether suicide risk is predicted by a summary measure of common mental distress (CMD (the p factor)) as well as by conventional psychopathological domains; to define the distribution of suicide risks over the population range of CMD; to test whether such distress mediates the medium-term persistence of suicide risks.

**Design** Two independent population-based cohorts.

**Setting** Population based in two UK centres.

**Participants** Volunteers aged 14–24 years recruited from primary healthcare registers, schools and colleges, with advertisements to complete quotas in age-sex-strata. Cohort 1 is the Neuroscience in Psychiatry Network (n=2403); cohort 2 is the ROOTS sample (n=1074).

**Primary outcome measures** Suicidal thoughts (ST) and non-suicidal self-injury (NSSI).

**Results** We calculated a CMD score using confirmatory bifactor analysis and then used logistic regressions to determine adjusted associations between risks and CMD; curve fitting was used to examine the relative prevalence of STs and NSSI over the population distribution of CMD. We found a dose–response relationship between levels of CMD and risk of suicide. The majority of all subjects experiencing ST and NSSI (78% and 76% in cohort 1, and 66% and 71% in cohort 2) had CMD scores no more than 2 SDs above the population mean; higher scores indicated the highest risk but were, by definition, infrequent. Pathway mediation models showed that CMD mediated the longitudinal course of both ST and NSSI.

**Conclusions** NSSI and ST in youths reflect CMD that also mediates their persistence. Universal prevention strategies reducing levels of CMD in the whole population without recourse to screening or measurement may prevent more suicides than approaches targeting youths with the most severe distress or with psychiatric disorders.

### Strengths and limitations of this study

► The samples were population based with several self-reported outcomes regarding suicidal risk.
► Replication of the findings in two independent cohorts strengthens confidence in the findings.
► Results were robust across different statistical models and approaches to data classification.
► Sample attrition was a limitation in both cohorts.
► Multiple imputations mitigated biases arising from attrition.

## INTRODUCTION

Adolescence sees the onset of a range of psychopathology including suicidal thoughts (ST) and non-suicidal self-injury (NSSI)[1–3] that individually or together convey heightened risk of suicide attempts.[4–6] Non-suicidal and suicidal self-harm predict completed suicide,[7] the second most common cause of deaths among 10–24 year-olds worldwide.[8] Moreover, ST and NSSI are significant problems in their own right, representing a considerable burden to individuals, their families and health services. Prediction and prevention of self-harm and suicide in young people are priorities but NSSI (5%–42% in community samples)[9 10] and ST (15%–25% in community samples)[11 12] are common so it is difficult to predict who will ultimately make a serious attempt[13] or die by suicide. Indeed, the usefulness of clinical risk protocols relying on the identification of a psychiatric diagnosis is questionable.[14 15] The same problems

affect public health suicide prevention programmes. A seminal study revealed a high prevalence of false negatives in prospective identification of suicide.[16] Prevention policies that embrace the whole population might overcome these difficulties but lack theoretical or empirical foundations.[1]

STs and behaviours are routinely considered as markers of depression (eg, in Diagnostic and Statistical Manual of Mental Disorders, Fifth Edition (DSM-5)) but by no means all young people dying by suicide have had a mood disorder.[17] NSSI is strongly associated with the risk of suicide when occurring in combination with any internalising or externalising symptoms,[18 19] or with any psychiatric diagnosis,[20] particularly multiple diagnoses.[21] Thus, this risk might be better predicted by multiple symptoms rather than by the presence of a single disorder, such as depression.

Recent studies suggest that a broad range of symptoms conventionally seen as components of distinct disorders are better construed as manifestations of a single, latent dimension distributed within the general population. This dimension has been variously referred to as the p factor,[22] general psychopathology[23] or, as we prefer here, common mental distress (CMD).[24 25] Parsimonious statistical models with dimensions that encompass low-prevalence phenomena such as psychotic experiences, fit empirical data better than models with distinct disorders.[22 26] High comorbidity of psychiatric diagnoses, shared causal factors and treatments, and transdiagnostic psychological and neural correlates support the validity of a CMD concept.[22–24 26–29] Suicide risk is related to multiple symptoms or disorders (and thus to higher CMD scores), not the presence of one specific symptom or disorder, so it is important to understand the nature of dose–response relationships between CMD and suicide risks. This could guide a clinical response in the face of suicide risk[30] and also shape population-based suicide prevention.

In this study, we describe the presence of a CMD dimension in young people aged 14–26 years and the occurrence of ST and NSSI referred to collectively, hereafter, as a suicide risk. We draw on a psychometric study[25] that demonstrated high theoretical validity and high measurement qualities of the CMD factor comprising measures of common mental illness (depression, anxiety, psychotic experiences, obsessions and compulsions) as well as traits and characteristics commonly considered to contribute to the general level of mental health (antisocial trait, well-being, self-esteem). Our approach had three steps whereby we:

1. Tested associations between CMD and suicide risk, and contrasted CMD with specific psychopathological domains, exploring the utility of this summary measure.
2. Defined the prevalence and relative risk of NSSI and ST across the distribution of CMD.
3. Established whether the $CMD_{T2}$ dimension measured at time 2 mediates the relationship between $ST_{T1}$ and $NSSI_{T1}$ at time 1 and $NSSI_{T3}$ and $ST_{T3}$ at time 3.

We used data from two population-based cohorts with complementary designs and very similar measures. In step 2 we used cross-sectional data from cohort 1, time 1 (used as a discovery sample) and cohort 2 (used as a stepwise replication sample); in the third step we used three longitudinal waves of cohort 1 (see details in the Methods section).

## METHODS
### Study design and participants
#### Cohort 1
Participants in the Neuroscience in Psychiatry Network 2400 cohort[31] were recruited largely via postal invitations sent through general practitioners and schools in Cambridgeshire and Greater London, UK. Data collection were carried out in two research centres: University College London and the University of Cambridge between November 2012 and December 2016. Purposive sampling obtained at least 200 males and 200 females from the community in five age groups: 14–15, 16–17, 18–19, 20–21 and 22–24 years. Three data collections took place a year apart (T1-T3). At T1, 2403 individuals returned questionnaires (average age 18.9 years, SD=3.0; 54% females); at T2, 1815 returned questionnaires (76% response, average age 20.0 years, SD=3.1; 56% female), and 1245 at T3 (52% of baseline; average age 21.0 years, SD=3.1; 59% female).

#### Cohort 2
The ROOTS study[32] was used for replication of findings from cohort 1. Two-stage sampling involved random selection of 27 schools in Cambridgeshire, UK. Eighteen schools agreed to participate; invitations were sent to 14 year-olds randomly selected from class registers and to their parents; 1238 students participated in the initial data collection (55% female) (and further four data collection waves took place). Note that in the current analysis we used only the data from the third data sweep collected between February 2008 and December 2009, when participants were of average age 17.5 years, SD=0.3 (n=1074, 56% female; 87% of baseline sample), the closest age to T1 of cohort 1.

Both cohorts comprised predominantly white European (77% in cohort 1 and 87% in cohort 2) young people, consistent with the self-ascribed demographics of the two study populations. Written consent from participants aged 14 or 15 years was supplemented by written consent from their parent or legal guardian; older participants gave their own written consent.

### Measures
Sociodemographic information was collected using routine methods.[31 33] The Index of Multiple Deprivation (IMD), a summary measure of the socioeconomic status of participants' residential neighbourhood, is calculated from census information.[34] Questionnaires of mental illness and wellness are set out in table 1 and items are

**Table 1** Measures used in both cohorts

| Variables | Measures | Cohorts | |
|---|---|---|---|
| **Outcome variables** | | NSPN$_{T1-T3}$ (1) | ROOTS$_{age\ 17}$ (2) |
| Suicidal thoughts (ST) | One item from the MFQ[50]: I thought about killing myself. Responses were recoded into a binary format: no ST (original response option *Never*) and ST (original response options *Sometimes* or *Mostly* or *Always*). | × | × |
| Non-suicidal self-injury (NSSI) | One question from the Drug, Alcohol and Self-Injury (DASI)[25] questionnaire asking about engaging in self-injury without suicidal intent during the last month. Responses were recoded into a binary format indicating the occurrence of NSSI or lack thereof. | × | |
| | One question asking about the occurrence of lifetime NSSI (DASI)[25] | | × |
| **Predictors** | | | |
| Conduct problems | 11-item Antisocial Behaviour Questionnaire[25] | × | × |
| Anxiety | 28-item Revised Children's Manifest Anxiety Scale[51] | × | × |
| Depression | 29 items from the 33-item MFQ[50] (all items except for 4 items measuring suicidality) | | |
| Obsessions and compulsions | 11-item Revised Leyton Obsessional Inventory[52] | × | × |
| Psychotic-like experiences | 11 items selected from the 74-item Schizotypal Personality Questionnaire (SPQ)[53] | × | |
| | 11 items from the 20-item semistructured interview from the Diagnostic Interview Schedule for Children-IV[54] | | × |
| Self-esteem | 10-item Rosenberg Self-Esteem Questionnaire*[55] | × | × |
| Well-being | 14-item Warwick-Edinburgh Mental Well-Being Scale*[56] | × | × |
| Impulsivity | 15 items from the 30-item Barratt Impulsiveness Scale[57] selected based on exploratory factor analysis—loadings above 0.25 | × | |
| Antisocial traits | Total score from the 17-item Antisocial Process Screening Device (APSD)[58] | × | |
| Schizotypal traits | Total score from the 74-item Schizotypal Personality Questionnaire (SPQ)[53] | × | × |

*Scales were reversely scored, thus higher scores indicated lower self-esteem and well-being; for all other measures higher score indicates more psychopathology.
MFQ, Mood and Feelings Questionnaire; NSPN, Neuroscience in Psychiatry Network.

listed in the online supplementary table 1. Scores in questionnaires were computed according to published manuals or validation studies (cited in table 1), standardised to unify their measurement scales.

### Statistical analysis

Confirmatory bifactor analysis with a weighted least squares mean and variance adjusted estimator in Mplus V.7.4 was used to compute factor scores for CMD in the three data sweeps of cohort 1 and cohort 2 based on the model validated elsewhere[25] (see CMD measures in table 1; the list of used items and details of bifactor modelling can be found in the online supplementary table 1). CMD factor scores were then used in all subsequent computations. Next, we addressed attrition in cohort 1 by means of multiple imputations (see details in the online supplementary material).

To prove that NSSI and ST were predicted by multiple psychopathological domains and also by CMD (which represents a summary of those domains), we used Stata V.12 to compute for cohort $1_{T1}$ and cohort 2 data sensitivity/specificity indicator—the area under the curve (reported in the online supplementary table 2) for NSSI and ST as criteria. We computed a series of logistic regressions, estimating ORs with CIs for each predictor (treated as categorical with the cut-off point above 1 SD and then continuous), while we controlled for effects of age and sex (figure 1).

For step 2, distributions of CMD scores in both cohorts were plotted against lines representing percentages of subjects reporting NSSI and ST within bands of CMD expressed as SDs (upper panel of figure 2) and against bar histograms representing NSSI and ST frequencies in both cohorts (lower panel of figure 2). In addition, NSSI and ST information curves were computed to determine in what range of the CMD dimension these items are located (see online supplementary figure 1).

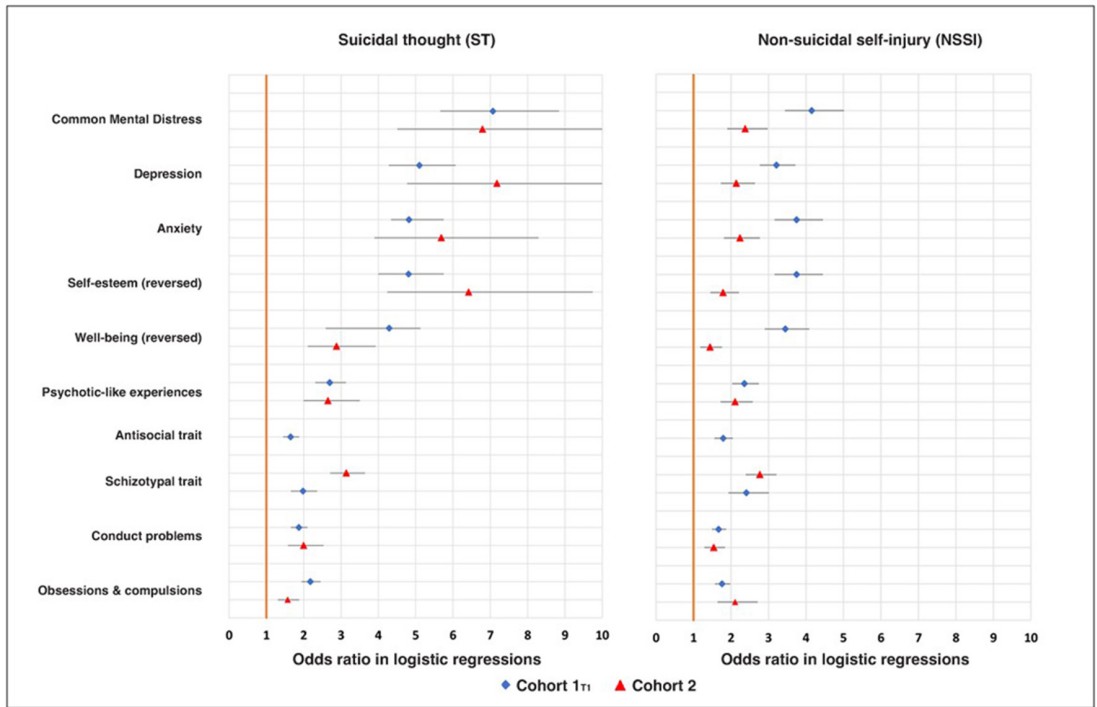

**Figure 1** OR in logistic regressions for suicidal thoughts (ST) and non-suicidal self-injury (NSSI) as outcomes predicted by psychopathological predictors (listed on the left) here treated as continuous variables; regressions were computed separately for each predictor and effects of age and sex were controlled in each regression for in both cohorts (see online supplementary table 2).

Using cohort $1_{T1-T3}$ data for step 3, we examined the longitudinal relationship between CMD, NSSI and ST (in particular the predictive role of CMD in persistence of NSSI and ST): we computed direct and mediation (via $CMD_{T2}$) effects of $ST_{T1}$ and $NSSI_{T1}$ on $NSSI_{T3}$ and $ST_{T3}$ in a pathway mediation model with CIs in Mplus V.7.4 (computing bias-corrected bootstrapping was not possible due to the use of multiply imputed data sets). We computed this model for the total sample (figure 3) and then for both sexes separately (online supplementary figure 2) using the multiple group method, so as to test a moderated mediation model (with $CMD_{T2}$ as a mediator, and sex as a moderator). Age was a control variable. In both pathway analyses $CMD_{T2}$ factor scores (computed on imputed data, as described above) were modelled as observed variables.

### Patient and public involvement
Patients and/or the public were not involved in the design, or conduct, or reporting, or dissemination plans of this research.

### RESULTS
### Step 1: associations of NSSI and ST with demographic and psychopathological variables
In both cohorts NSSI and ST were unrelated to demographic variables, including sex and age (see online supplementary tables 3 and 4); CMD was negatively related to male gender (online supplementary table 5). When examined descriptively over the pooled age groups, the prevalence of NSSI and ST mirrored the CMD levels (see online supplementary figure 3). CMD and all 'conventional' mental health disorders predicted NSSI and ST (ie, had statistically significant ORs in logistic regression models—see figure 1 and online supplementary table 2).

### Prevalence of NSSI and ST in the two cohorts
In cohort 1 (n=2403) there was no statistically significant change in the prevalence of NSSI (within the last month) over the three time points: in the imputed data 9.3% (n=223) reported $NSSI_{T1}$, 8.3% (n=199) $NSSI_{T2}$ and 8.2% (n=197) $NSSI_{T3}$. Similarly, there was no statistically significant change in prevalence of ST (within the last 2 weeks) over the three time points: 10.1% (n=243) $ST_{T1}$, 11.4% (n=274) $ST_{T2}$ and 11.7% (n=281) $ST_{T3}$ (see online supplementary tables 6 and 7).

In cohort 2 (n=1074), 11.7% (n=126) reported lifetime NSSI and 5.4% (n=58) reported ST within the last 2 weeks. Accuracy and precision of these prevalence estimates were affected by attrition (see the Discussion section: *Limitations*). Attrition in cohort 1 at T2 and T3 was only marginally related to demographic and exposure variables at T1 (Spearman's r=0.05–0.12), but unrelated to the outcome—NSSI and ST (see online supplementary table 8).

### Step 2: associations of NSSI and ST with CMD
Next, we focused on absolute risk (Absolute risk is the probability or chance of an event. It is usually used for

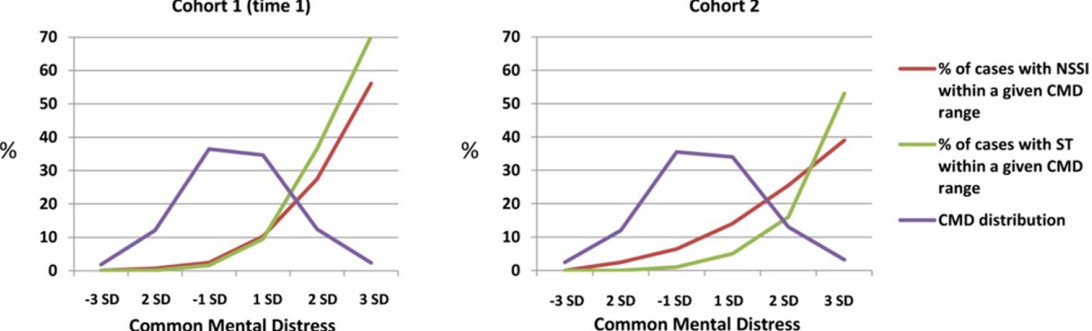

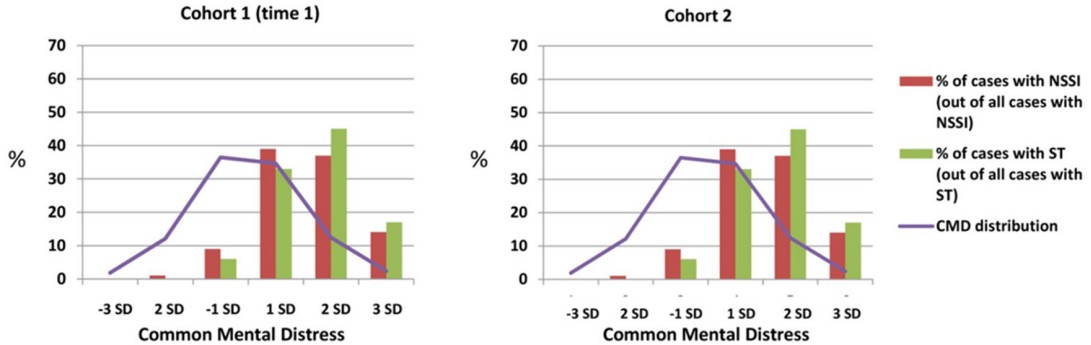

**Figure 2** Upper panel shows the dose–response effect of common mental distress on non-suicidal self-injury (NSSI) and suicidal thought (ST) in cohort 1 and cohort 2. The lower panel shows the proportion of total reports in NSSI and ST broken down by SDs of common mental distress; these add up to 100% from left to right. The normal population distribution of CMD, which was strikingly similar, but not identical, in cohorts 1 and 2, is shown by the purple line (see density plots in online supplementary figure 1). CMD, common mental distress.

the number of events (eg, a suicide) that occurred in a group, divided by the number of people in that group) and the numbers of NSSI and ST events generated by these risk functions. The dose–response curves in the upper panel of figure 3 show that relative risks (A relative risk compares the risk of a health event (eg, a suicide) among one group with the risk among another group) of NSSI and ST increased markedly with increasing severity

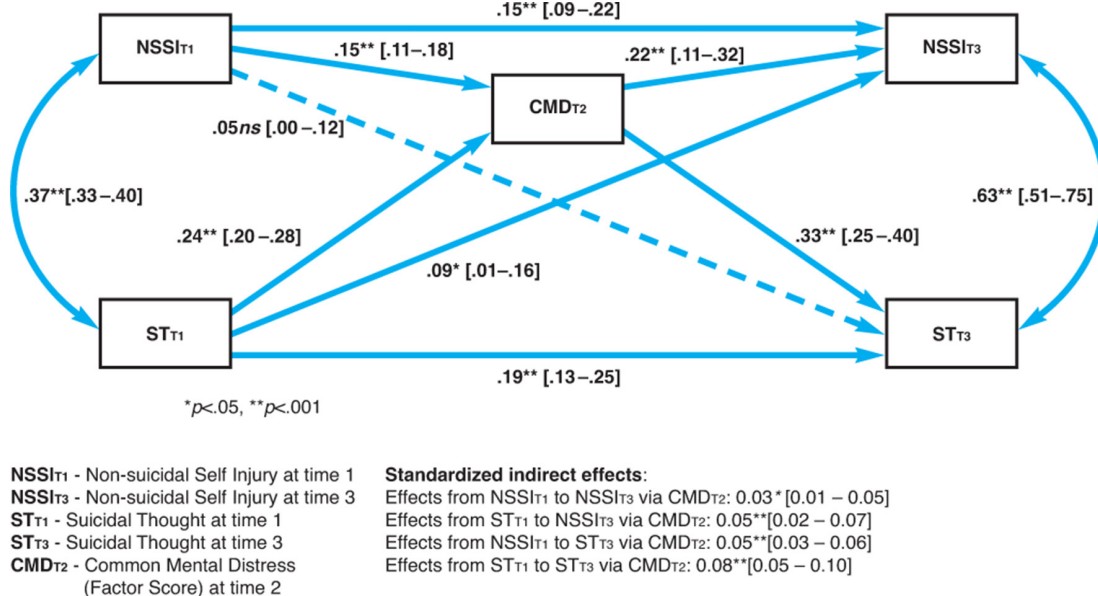

**Figure 3** Mediation effect of common mental distress at time 2 in cohort 2: standardised pathway coefficients with CIs in square brackets.

of CMD, the highest risks being in those with very high scores beyond 2 SDs above the mean. On the other hand, most participants from both cohorts who reported NSSI or ST had mild (1 SD above the mean) to moderate (2 SDs above the mean) CMD scores (lower panel of figure 3). CMD was normally distributed (see online supplementary figure 4) so these scores were much more common; only a minority of the total reports came from the few participants with very high CMD (>2 SDs above mean CMD). Thus, the majority of subjects experiencing ST or NSSI (cohort 1: 78% and 76%; cohort 2: 66% and 71%, respectively) had CMD scores within 2 SDs above the population mean. Very high CMD scores indicated the highest suicide risk but were rare, so generated the minority of events.

### Step 3: mediating effect of CMD on suicide risks in cohort 1 over time

Cohort 1 $CMD_{T2}$ contributed to the persistence of NSSI and ST over time (ie, $NSSI_{T1}$ predicted $NSSI_{T3}$ directly, and via mediation through $CMD_{T2}$; it also completely mediated the longitudinal effect of $NSSI_{T1}$ on $ST_{T3}$). Moreover, $CMD_{T2}$ contributed to the persistence of ST over time (ie, $ST_{T1}$ predicted $ST_{T3}$ directly, as well as via mediating variable—$CMD_{T2}$). Overall, $CMD_{T2}$ was a stronger predictor of $NSSI_{T3}$ and $ST_{T3}$ than the antecedent variables measured at T1 (see figure 3). The mediation effects of $CMD_{T2}$ were similar for boys and girls (ie, the effects were not moderated by sex—online supplementary figure 2 and online supplementary table 9). $Age_{T1}$ was not a significant predictor of any variable in the model; the results when age was controlled for were very similar to those without controlling for age.

### DISCUSSION

In the present study, all the domains of psychopathology and mental wellness available (depression, anxiety, self-esteem, well-being, psychotic-like experiences, antisocial trait, schizotypal trait, conduct problems, obsessions and compulsions) predicted risk of NSSI and STs. Thus, the CMD factor with a normal population distribution appeared a parsimonious and efficient summary of these domains and was, itself, a key predictor of suicide risk in both cohorts. NSSI and ST were not confined to participants scoring in the very high, quasiclinical range for CMD. Around half of all participants expressing NSSI or ST came from those scoring up to 1 SD above mean CMD in a dose–response manner. The majority expressing these phenomena (two-thirds to three-quarters) scored within 2 SD above the mean (figure 2).

Regarding medium-term determinants of persistent NSSI and ST we showed (figure 3) that $CMD_{T2}$ mediated the persistence of NSSI and ST over 2 years, independent of gender and age. This mediation operates in two stages: first, ST and NSSI persist because these behaviours are markers for worsening CMD in the general population. This extends findings in adolescents with depressive

disorder, where STs are a predictor of poor outcome.[35] Second, this greater CMD itself predicts the risk for further STs and behaviours.

### Strengths

Both cohorts were designed on epidemiological principles to capture behavioural and psychological variation in the population during the postpubertal epoch during which risk for psychopathology accelerates. Replication of the findings in these independent cohorts strengthens confidence in the findings, as does internal consistency between cross-sectional associations found in both cohorts, and longitudinal associations found in cohort 1.

### Limitations

Sample attrition was the main bias in both cohorts. Each retained more young women than men; we found marginally higher attrition among lower socioeconomic class, participants of non-white ethnicity and those with higher CMD (online supplementary table 8). Cohort 1 is robustly representative of the England and Wales population,[31] whereas cohort 2 under-represents participants with lowest socioeconomic status.[32] However, we have no reason to suppose that attrition biased our results, as it was unrelated to NSSI and ST (online supplementary table 8). If there was a bias, it probably limits power rather than skewing an effect and is mitigated by replication between the cohorts. We used multiple imputation to minimise this bias.

There was only modest reliability of our obsessionality measure and a skewed measure of conduct problems in cohort 1. A completely comprehensive range of psychopathological (and behavioural) items was unavailable; we did not have measures of unstable or abnormally elevated mood, addictions, eating disorders or hyperactivity. Thus, our measurement of CMD focused primarily on internalising rather than externalising symptoms. Future studies could include a broader range of measures and extend the investigation into clinical populations to improve measurement precision at the highest levels of CMD. Although ethnicity and socioeconomic status (indicated by IMD) were unrelated to ST and NSSI (online supplementary tables 3 and 4), and thus were not included in our analyses, we did not control for the effect of other possible confounders such as adverse life experiences, early trauma, family structure or more detailed information about family socioeconomic situation (unemployment, poverty, and so on). Finally, we could not account for the effects of clustered design in the modelling, due to unavailability of the information about clustering of participants in both cohorts.

Our findings provide yet more evidence that a latent mental distress factor, conceptually akin to the p factor, is a useful summary measure of psychopathology in the general population,[24] diagnostic[22] and clinical[23] samples. We speculate that psychopathological items accumulate in a probabilistic manner rather than in diagnostic clusters, with common phenomena concerning depression

and anxiety much more likely to occur before rarer phenomena such as NSSI, ST or psychotic experiences. Less frequent phenomena begin to co-occur as the severity of psychological disorder (or CMD) increases, in terms of more mental and behavioural phenomena or symptoms. This begins to yield clusters linked by common items that current diagnostic systems tend to ignore. This is consistent with the co-occurrence of suicidal risk and psychotic experiences seen in other[36–38] studies of young people, and with the present item response theory (IRT) analysis showing that NSSI and ST are measuring the higher end of CMD (online supplementary figure 1). The approach we have followed illustrates the value of moving away from categorical classification and embracing an empirically rooted, dimensional, hierarchical taxonomy in psychopathology research.[39] Such hierarchical approaches to phenomenological classification had been put forward before[40] or shortly after[41] the publication of DSM-3 and its successor classifications. Hierarchical models merit renewed interest,[42] as they may resolve problems of comorbidity[26] as well as overlapping causes and biological mechanisms for suicide risk and other phenomena.[43 44] In contrast to the CMD idea, there is also increasing interest in approaches focusing on individual symptoms and experiences, particularly to guide individual clinical interventions, rather than grouping the symptoms into diagnostic categories or higher order constructs.[45] Future studies may investigate and compare the utility of such novel approaches (CMD and item-focused approach) for clinical practice and public health policies.

Our findings also have major implications for intervention and prevention of STs and behaviours. Clinically, the results suggest that NSSI and ST should never be dismissed or downplayed when they occur in young people without clear evidence of psychiatric disorder, a logical fallacy because NSSI and ST are *themselves* indicators of higher distress on a CMD factor. NSSI and ST will usually, but not always occur with other, more common psychopathology and their co-occurrence is a strong risk factor for suicide attempts.[6] Thus, NSSI and ST merit a swift professional response regardless of whether or not they occur with other symptoms that take individuals beyond conventional clinical thresholds and trigger traditional clinical risk protocols. Our findings help explain why research focused on high-risk subjects has yet to translate into useful clinical prediction tools.[14 15 45]

From a public health and prevention perspective, the fact that rates of NSSI and ST begin to accelerate at levels of CMD well within a normal or non-clinical range argues strongly for universal interventions overtly aimed at lowering the population mean CMD and shifting the curve to the left. This should be alongside targeted approaches and effective clinical services.[46] Strategies concentrated on clinical populations, those with evidence of a psychiatric disorder or other individual markers will miss the majority of individuals experiencing ST or engaging in NSSI because there are so few compared with those at lower risk: the *prevention paradox*.[30]

Defining putative universal interventions to shift the population distribution of CMD will require careful research that can draw from other areas of medicine such as cardiovascular disease and stroke.[30] Elements have been widely scoped in the USA[15] and elsewhere, but not for constructs of population health and well-being such as CMD. Interventions may involve decreasing common triggers[15] or improving young people's abilities to cope with stressors[47–49]

**Author affiliations**
[1]Psychiatry, University of Cambridge, Cambridge, UK
[2]Psychology, University College Dublin, Dublin, Ireland
[3]Psychology, University of Bath, Bath, UK
[4]Wellcome Centre for Human Neuroimaging, University College London, London, UK
[5]Max Planck University College London Centre for Computational Psychiatry and Ageing Research, University College, London, UK
[6]Research Department of Clinical, Educational and Health Psychology, University College London, London, UK
[7]NIHR Applied Research Collaboration East of England, Cambridge, Cambridgeshire, UK

**Acknowledgements** The work has been carried out in the Department of Psychiatry, University of Cambridge. We thank the NSPN and ROOTS participants and Dr Golam Khandaker for his comments and NSPN Consortium (see the list of members in the online supplementary material).

**Contributors** EP conceptualised the study, computed statistical analyses and drafted the manuscript. PJ provided senior supervision, conceptualised the study, advised on statistical analyses, read and critically appraised the manuscript, redrafted and edited the manuscript. JS provided statistical advice, replicated multiple imputations, provided data from multiple imputations, read and critically appraised the manuscript. SASN advised on handling the missing data, replicated multiple imputations, read and critically appraised the manuscript. PW, RD, IG, ETB and PF read and critically appraised the manuscript, and provided key referred articles. MSC contributed to data collection and project management, and provided advice on bifactor modelling. GP contributed to data collection and project management.

**Funding** The ROOTS study was supported by a Wellcome Trust Grant (Grant No 074296) to IG and PJ, the NIHR Collaborations for Leadership in Applied Research and Care (CLAHRC) East of England, and the NIHR Cambridge Biomedical Research Centre. The NSPN study was supported by the Wellcome Trust Strategic Award (095844/Z/11/Z) to IG, ETB, PJ, RD and PF.

**Disclaimer** The funding sources had no role in the design and conduct of the study; collection, management, analysis and interpretation of data; preparation, review or approval of the manuscript; and decision to submit the manuscript for publication.

**Competing interests** ETB and PF are in receipt of National Institute for Health Research (NIHR) Senior Investigator Awards (NF-SI-0514-10157 and NF-SI-0514-10117). PF was in part supported by the NIHR Collaboration for Leadership in Applied Health Research and Care (CLAHRC) North Thames at Barts Health NHS Trust. PW has recent/current grant support from NIHR, Cambridgeshire County Council and CLAHRC East of England. PW discloses consulting for Lundbeck and Takeda. PJ is supported by the NIHR Applied Research Collaboration East of England and discloses consulting for Janssen and Ricordati. At the time of the study ETB was employed half time by the University of Cambridge and half time by GlaxoSmithKline in which he holds stock. EP, SASN, IG and JS have no competing interests.

**Patient consent for publication** Not required.

**Ethics approval** Ethical approval was obtained for cohort 1 from the National Health Service Research Ethics Service (No 97546) and for cohort 2 from the Cambridgeshire 2 REC (No 03/302).

**Provenance and peer review** Not commissioned; externally peer reviewed.

**Data availability statement** Data are available in a public, open access repository. Data are available upon reasonable request. EP had full access to all the data in the study and takes responsibility for the integrity of the data and the accuracy

of the data analysis. The data are deposited in the University of Cambridge Data Repository, with the placeholder DOI https://www.repository.cam.ac.uk/handle/1810/278070 available to researchers via openNSPN@medschl.cam.ac.uk.

**ORCID iD**
Ela Polek http://orcid.org/0000-0003-1405-2269

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
