## [Reviewer comments · BMJ Open]

ARTICLE DETAILS

TITLE (PROVISIONAL)	How do the prevalence and relative risk of non-suicidal self-injury and suicidal thoughts vary across the population distribution of common mental distress (the p-factor)? Observational analyses replicated in two independent UK cohorts of young people
AUTHORS	Polek, Ela; Neufeld, Sharon A. S.; Wilkinson, Paul; Goodyer, Ian; St Clair, Michelle; Prabhu, Gita; Dolan, Ray; Bullmore, Edward; Fonagy, Peter; Stochl, Jan; Jones, Peter

VERSION 1 – REVIEW

REVIEWER	Peter Taylor University of Manchester, UK
REVIEW RETURNED	23-Jul-2019

GENERAL COMMENTS	I think this is an interesting and competently conducted study on the association between a singular common mental distress factor and suicide risk. The use of large existing datasets is positive. I think the paper provides an interesting counter-point to arguments within the field to be more focused on individual symptoms, but there is a lack of engagement with this, which I think is a missed opportunity. There are various possible issues I outline below: Abstract: Abbreviations NSSI and ST need writing in full first time they are used. Introduction: The problem with identifying who will go on to attempt suicide based on NSSI and suicidal thoughts is correctly identified. However, despite this, NSSI and suicidal thoughts are still used to represent “suicide risk”. I think more of a defence is needed around why NSSI and suicidal thoughts are still relevant outcomes, given their limited predictive value. It would also be helpful to move this justification beyond just suicide risk to consider the wider issues presented by NSSI and ST; these are problems in their own right, that can represent a considerable burden to individuals and their families, as well as health services. In contrast to the CMD idea, there is also a push to focus more on individual symptoms and experiences, rather than to groups these into higher-order constructs (e.g. “Depression sum-scores don’t add up: why analyzing specific depression symptoms is essential” https://bmcmmedicine.biomedcentral.com/articles/10.1186/s12916-015-0325-4). I think the CMD approach needs further justification in light of these arguments.
---

The term “mediate” is used in the aims but it is not immediately clear what is being tested. I think a clearer statement is needed here, clarifying that the model involves the relationship between earlier and later suicide risk being mediated by CMD.

Method:

I am unclear about the need to switch the analysis from MPlus and a latent variable modelling (LVM) framework, to STATA and a simpler logistic regression framework. When CMD was analysed in STATA how was this variable captured? Was it modelled still as a latent variable, or were factor scores or some other method used? It would be better perhaps to stick with the LVM framework throughout, so that CMD can be effectively modelled as a latent variable (reducing measurement error) when predicting suicide risk. I wondered about the decision to use multiple imputation, rather than allowing the model to manage missing data within MPlus. Mplus can provide estimation of model parameters where there is incomplete data for endogenous variables. My understanding is that this approach tends to fare as well as multiple imputation in terms of bias.

Regarding the bifactor LVM – I am not sure if multi-group CFA can be used for the same sample across multiple time-points. Normally this sort of approach assumes the groups represent independent samples, which is not the case here (the data is essentially nested, time-points within participants). I am not sure if it is appropriate to test for invariance across time-points in the same way you would between independent samples. Some explanation is needed as to how this is being managed.

The validity of the general CMD factor is a major aspect of this study. It is possibly an issue that no alternative models are compared against the CMD model. Comparing the CMD factor model against a range of plausible alternatives would provide further evidence that a single CMD factor is justifiable. I note some of the factor loadings are actually quite low (e.g. below .5) and I wonder whether treating these difficulties as a distinct but correlated set of factors would ultimately be a better solution.

Results

I don't think it is accurate to say that “CMD and all conventional psychopathological predictors of NSSI and ST had statistically significant and similar size ORs in logistic regression models”. The ORs actually vary quite widely given what is typical for these sorts of associations (e.g. from 7 to 1.6)

In the path diagram of the mediation model are the numbers for standardized or non-standardized effects (I can see standardized indirect effects are reported, but what about the other values)? Notably, I think non-standardized effects should also be reported for all parameters including indirect effects. Within this analysis of indirect effects, is CMD being estimated as a latent variable?

There is inadequate information about the IRT analysis, what the purpose of this is and what it is demonstrating over and above the other analyses. Either drop this analysis or provide more clarity I think.

Discussion

The argument being made in the opening sentence is a little unclear, I do not think it is suggested by researchers or clinicians

	that depression is the only form of psychopathology linked to suicide risk, so I am not sure what point is being made here. I think more caution is also needed in the claims made by the study. The data is correlational, and analyses do not seem to adjust for pre-existing difficulties (e.g. the mediation analysis does not adjust for pre-existing problems with NSSI and ST as far as I can see). Given this, the analyses are limited in what they can say about the directionality of these associations and of course do not allow any inference about causal relationships. I think this needs explicitly stating. Moreover, the language needs revising in places to avoid suggesting causality, (e.g., “greater CMD, itself increases the risk for further suicidal thoughts and behaviours”). Similarly, I think claims about the usefulness of the CMD factor should be presented with more caution. The results suggest a single CMD factor does fit the available data, and as would be expected, is correlated with suicide risk. However, there is no direct comparison with alternative ways of representing mental distress. I also wondered about the usefulness of a singular CMD factor from a clinical perspective. The OR for CMD were similar to just depression alone, and it would be much quicker as a clinician to measure just depression rather than CMD more broadly, and possibly without much loss of relevant information about risk. Is there any argument that can be made for the clinical benefits of a singular CMD factor?
--	--

REVIEWER	Keely Cheslack-Postava New York State Psychiatric Institute/Columbia University USA
REVIEW RETURNED	08-Aug-2019

GENERAL COMMENTS	This manuscript examines the relationship between “Common Mental Distress”, a latent dimension of general psychopathology, defined using a factor analytic model, with suicidal thoughts and non-suicidal self-injury in 2 cohorts of adolescents and young adults. This study has potential important implications for prevention of suicide-related behaviors at a population level. The strengths include a very thorough analysis and documentation, replication (of part of the study) in 2 separate populations, an extensive list of questionnaire items contributing the definition of CMD, and mediation analysis with longitudinal data measured over 3 timepoints. I have a few main concerns and some additional comments. 1. Models (i.e. association of CMD with ST and NSSI; mediation models) were adjusted for only a very limited set of covariates (i.e. age and sex)... what about SES, family structure and environment, history of adverse events or trauma, to name only a few?? Of note, when examining mediation, not only confounding of the exposure-outcome association is of concern, but of the exposure-mediator and mediator-outcome associations. In the mediation model, couldn't the baseline (i.e. T1) CMD confound the described associations? It seems a strength of this longitudinal data would be the ability to examine the effects of changes over time in CMD levels (i.e. increase from T1 to T2).
---

	2. P. 12, the first paragraph under “Implications and Conclusions” could use overall re-writing for clarity, in particular “psychopathology is generated in a probabilistic manner...” (generated how?); “this begins to yield clusters ...” (of what?); explain briefly “hierarchical approaches”. 3. P. 14, last paragraph. This paragraph seems to delve into detail beyond what is supported by this study and a briefer treatment would seem appropriate (i.e. the evidence presented here suggests the conceptual impact of interventions to shift population distribution of CMD, but does not apply to anything about specific potential interventions, i.e. by digital platforms or social media). 4. Many items go into defining the measure of CMD, how does that affect its potential utility? Minor  1. Abstract, line 26 states “Volunteers age 14-24 years” but the intro section (p. 5) says 14-26 years. 2. Abstract, abbreviations ST and NSSI should be spelled out. 3. Introduction, p. 5, lines 33-47. The statement of the aims comes across a bit disjointedly – first, “... we aimed to test here associations between CMD and suicide risk,” And then a numbered list of 2 additional questions. It would be easier to follow both here and in the results section if all of the aims could be presented as one cohesive list (either numbered or narratively). 4. P. 7, line 40, spell out “WLMSV” on first use.
--	---

REVIEWER	Karen H. Larwin Youngstown State University
REVIEW RETURNED	17-Sep-2019

GENERAL COMMENTS	Thank you for the opportunity to read and review this important manuscript about Common Mental Distress as a potential indicator of suicide. I have a few considerations for revision. First, please indicate the full term prior to using the acronym. For example with "NSSI" and "ST" in the abstract. The reader might not know what you are referencing. Second, I believe that in addition to the recommendation to provide interventions to help lower the CMD experienced by these individuals, I would consider adding some discussion about why the CMD might be where it is at and how these individuals might benefit from some greater understanding of lifes ups and downs, reflections on values, or work on mindfulness. Also, I would like you to comment on whether order-effect was a limitation with the items used. Lastly, did you conduct reliability estimates on any of the data? This information should be provided so that we can understand how well these items worked across the two groups. These are all minor revisions and should be easily managed.
--

VERSION 1 – AUTHOR RESPONSE

Reviewer(s)' Comments to Author:

Reviewer: 1

Reviewer Name: Peter Taylor

Institution and Country: University of Manchester, UK

Please state any competing interests or state 'None declared': None

Please leave your comments for the authors below

I think this is an interesting and competently conducted study on the association between a singular common mental distress factor and suicide risk. The use of large existing datasets is positive. I think the paper provides an interesting counter-point to arguments within the field to be more focused on individual symptoms, but there is a lack of engagement with this, which I think is a missed opportunity. There are various possible issues I outline below:

Abstract:

Abbreviations NSSI and ST need writing in full first time they are used.

Done (page 2)

Introduction:

The problem with identifying who will go on to attempt suicide based on NSSI and suicidal thoughts is correctly identified. However, despite this, NSSI and suicidal thoughts are still used to represent “suicide risk”. I think more of a defence is needed around why NSSI and suicidal thoughts are still relevant outcomes, given their limited predictive value. It would also be helpful to move this justification beyond just suicide risk to consider the wider issues presented by NSSI and ST; these are problems in their own right, that can represent a considerable burden to individuals and their families, as well as health services.

We added in the Introduction the following sentence (page 4):

Moreover, ST and NSSI are significant problems in their own right, representing a considerable burden to individuals, their families and health services.

In contrast to the CMD idea, there is also a push to focus more on individual symptoms and experiences, rather than to group these into higher-order constructs (e.g. “Depression sum-scores don’t add up: why analyzing specific depression symptoms is essential” <https://bmcmedicine.biomedcentral.com/articles/10.1186/s12916-015-0325-4>). I think the CMD approach needs further justification in light of these arguments.

We do acknowledge the importance of the approaches focusing on items or symptoms (as opposed to diagnostic categories) and have taken this on, to a degree, in our Step 1 as reported. However, detailed reporting on an item-level analysis is beyond the scope of this study, which includes a broad range of other analyses (as the reviewer noted). We suggested that comparative studies examining the utility of item-level and CMD approaches might be carried out in the future (on page 13) and we added the suggested reference to the bibliography on page 22 (number 45):

In contrast to the CMD idea, there is also increasing interest in approaches focusing on individual symptoms and experiences, particularly to guide individual clinical interventions, rather than grouping the symptoms into diagnostic categories or higher-order constructs⁴⁵. Future studies may investigate and compare the utility of such novel approaches (CMD and item-focused approach) for clinical practice and public health policies.

The term “mediate” is used in the aims but it is not immediately clear what is being tested. I think a clearer statement is needed here, clarifying that the model involves the relationship between earlier and later suicide risk being mediated by CMD.

Aim 2 has been rephrased to read (page 5):

Does the CMD dimension mediate the relationship between ST_{T1} and $NSSI_{T1}$ at time 1 and $NSSI_{T3}$ and ST_{T3} at time 3?

Method:

I am unclear about the need to switch the analysis from MPlus and a latent variable modelling (LVM) framework, to STATA and a simpler logistic regression framework. When CMD was analysed in STATA how was this variable captured? Was it modelled still as a latent variable, or were factor scores or some other method used?

We computed factor scores in bifactor model and then used them in all subsequent analyses. We added the sentence (here in bold) in the paragraph *Statistical analysis* to clarify this (page 7):

“Confirmatory bifactor analysis with a weighted least square mean and variance adjusted (WLMSV) estimator in Mplus 7.4 was used to compute factor scores for CMD in the three data sweeps of Cohort 1 and Cohort 2 based on the model validated elsewhere²⁵ (see CMD measures in Table 1 beneath; the list of used items and details of bifactor modelling can be found in Supplementary table 1). **CMD factor scores were then used in all subsequent computations.** Next, we addressed attrition in Cohort 1 by means of multiple imputations (see details in the Supplement).”

It would be better perhaps to stick with the LVM framework throughout, so that CMD can be effectively modelled as a latent variable (reducing measurement error) when predicting suicide risk.

We have not amended the manuscript but respond here:

We were interested in generating population distributions of CMD scores, not only obtaining estimates in regression or SEM models, thus factor scores of CMD were generated first and then used in subsequent analyses. This approach was also dictated by the necessity to binarize predictors (including CMD) at 1SD in logistic models predicting NSSI and ST.

I wondered about the decision to use multiple imputation, rather than allowing the model to manage missing data within Mplus. Mplus can provide estimation of model parameters where there is incomplete data for endogenous variables. My understanding is that this approach tends to fare as well as multiple imputation in terms of bias.

We have not amended the manuscript but respond here:

Indeed, FIML estimator in Mplus allows handling data missing at random while using raw data, and Mplus developers advise that FIML is “asymptotically equivalent” to MI (<http://www.statmodel.com/discussion/messages/22/2440.html?1425099486>). MI has the advantage that all variables in the dataset (and not only those in the model, as in the FIML), contribute to MAR assumption and can be used to generate imputed data.

Regarding the bifactor LVM – I am not sure if multi-group CFA can be used for the same sample across multiple time-points. Normally this sort of approach assumes the groups represent independent samples, which is not the case here (the data is essentially nested, time-points within participants). I am not sure if it is appropriate to test for invariance across time-points in the same way you would between independent samples. Some explanation is needed as to how this is being managed.

We have not amended the manuscript but respond here:

Multiple group method is a broadly used method for testing longitudinal measurement invariance, see e.g.:

Kim, E. S., & Yoon, M. (2011). Testing measurement invariance: A comparison of multiple-group categorical CFA and IRT. *Structural Equation Modeling*, 18, 212–228.

Muthén, B. O., & Asparouhov, T. (2002). Latent variable analysis with categorical outcomes: Multiple-group and growth modeling in Mplus (Mplus Web Notes No.4). Retrieved from <http://www.statmodel.com/downloads/webnotes/CatMGLong.pdf>

In order to test longitudinal measurement invariance Mplus developers suggest setting factor loadings and thresholds to be equal over time (i.e., at each data wave) and checking if holding them equal across time points (as opposed to estimating them freely in each data wave) affects the overall fit of the MGM model (see chapter 14, p. 433 of Mplus User's Guide). We followed this procedure.

The validity of the general CMD factor is a major aspect of this study. It is possibly an issue that no alternative models are compared against the CMD model. Comparing the CMD factor model against a range of plausible alternatives would provide further evidence that a single CMD factor is justifiable. I note some of the factor loadings are actually quite low (e.g. below .5) and I wonder whether treating these difficulties as a distinct but correlated set of factors would ultimately be a better solution.

We have not amended the manuscript but respond here:

We are clear in the introduction that we draw on existing psychometric work. The original psychometric study of St Claire et al. (2017), which our bifactor model is based upon, evaluated and compared a range of models, including single-factor, correlated-factor, second-order factor, and bi-factor models. Fit indices found for the bi-factor model were superior as compared to fit indices of other models (see Table 1 in St Claire et al. (2017)).

Ref:

St Clair, C. M., Neufeld, S., Jones, B.P., et al. Characterising the latent structure and organisation of self-reported thoughts, feelings and behaviours in adolescents and young adults. *PLOS One*. 2017; 12(4), 1-27. doi: <https://doi.org/10.1371/journal.pone.0175381>

Results

I don't think it is accurate to say that "CMD and all conventional psychopathological predictors of NSSI and ST had statistically significant and similar size ORs in logistic regression models". The ORs actually vary quite widely given what is typical for these sorts of associations (e.g. from 7 to 1.6)

The phrase "and similar size" was removed from this sentence (page 9).

In the path diagram of the mediation model are the numbers for standardized or non-standardized effects (I can see standardized indirect effects are reported, but what about the other values)? Notably, I think non-standardized effects should also be reported for all parameters including indirect effects. Within this analysis of indirect effects, is CMD being estimated as a latent variable?

We added Supplementary Table 9 to the online Supplement (page 13 and 14) reporting the results of the pathway analysis in a female, male and a total sample (direct and indirect effects, standardised and non-standardised coefficients, standard errors for coefficients and 95% confidence intervals). We also added the following sentence to the manuscript (page 8): "In both

pathway analyses CMD_{T2} factor scores (computed on imputed data, as described above) were modelled as observed variables.” to clarify how CMD was treated in mediation analyses.

There is inadequate information about the IRT analysis, what the purpose of this is and what it is demonstrating over and above the other analyses. Either drop this analysis or provide more clarity I think.

We added (here in bold) the following to the online supplement (page 15, Supplementary figure 1):

Item Response Theory (IRT) analysis is concerned, broadly speaking, with investigating the relationship between items and the latent construct. Here we computed item response function showing how much information NSSI and ST (here treated as indicators of CMD) contribute to the latent variable – CMD. The above graph shows that NSSI and ST provided information in above-average to high ranges of CMD, with the peak of the information curves for NSSI occurring around +2 SD in both cohorts. The information curve for ST in Cohort 2 was flatter, suggesting less contribution to the latent CMD dimension than ST had in Cohort 1_{T1} dataset. This may be due to the differences in age structure and psychopathology status in both cohorts. Nonetheless, in both cohorts the peak in the ST curves occurred between +2 and +3 SD (high end of the CMD dimension), showing that ST lies on the more severe spectrum of CMD dimension than NSSI does.

Discussion

The argument being made in the opening sentence is a little unclear, I do not think it is suggested by researchers or clinicians that depression is the only form of psychopathology linked to suicide risk, so I am not sure what point is being made here. I think more caution is also needed in the claims made by the study. The data is correlational, and analyses do not seem to adjust for pre-existing difficulties (e.g. the mediation analysis does not adjust for pre-existing problems with NSSI and ST as far as I can see). Given this, the analyses are limited in what they can say about the directionality of these associations and of course do not allow any inference about causal relationships. I think this needs explicitly stating.

We revised the opening paragraph in the Discussion section to read (page 11):

In the present study, all included conventional domains of psychopathology and mental wellness (depression, anxiety, self-esteem, well-being, psychotic-like experiences, antisocial trait, schizotypal trait, conduct problems, obsessions and compulsions) predicted risk of non-suicidal self-injury (NSSI) and suicidal thoughts (ST). Thus, the common mental distress factor with a normal population distribution appeared as a parsimonious and efficient summary of these domains and was, itself, a key predictor of suicide risk in both cohorts.

Also, we deleted from the Discussion (Limitations of the study paragraph), the phrase (page 12): “We broadened our scope far beyond depression, usually the focus of psychological disturbance in suicidality research”.

Moreover, the language needs revising in places to avoid suggesting causality, (e.g., “greater CMD, itself increases the risk for further suicidal thoughts and behaviours”).

We changed the word “increases” (which, indeed, may imply causality) on pages 4 and 11:

“NSSI is strongly associated with the risk of suicide when occurring in combination with any internalising or externalising symptoms^{18,19}, or with any psychiatric diagnosis²⁰, particularly multiple diagnoses²¹.”

“Second, this greater CMD, itself, predicts the risk for further suicidal thoughts and behaviours.”

Similarly, I think claims about the usefulness of the CMD factor should be presented with more caution. The results suggest a single CMD factor does fit the available data, and as would be expected, is correlated with suicide risk. However, there is no direct comparison with alternative ways of representing mental distress.

We have not amended the manuscript but respond here:

In our step 1 we carried out direct comparison of the utility of CMD with other “conventional” disorders (e.g., depression, anxiety, obsessionality, schizotypal traits, antisocial traits).

I also wondered about the usefulness of a singular CMD factor from a clinical perspective. The OR for CMD were similar to just depression alone, and it would be much quicker as a clinician to measure just depression rather than CMD more broadly, and possibly without much loss of relevant information about risk. Is there any argument that can be made for the clinical benefits of a singular CMD factor?

We have not amended the manuscript but respond here:

We should re-iterate here the main Public Health-facing message of the paper, which was emphasising the need to go beyond the “traditional” focus on the clinical population in suicide prevention approaches. We propose that in order to effectively lower suicide risk on the population level, a population approach is needed aimed at lowering common mental distress rather than focusing exclusively on individuals with very high mental distress (i.e., those who make the clinical population). It was not our aim to demonstrate the usefulness of the CMD in the clinical practice, even though we did mention clinical implications such as measuring the risk with a greater precision or going beyond the focus on depression-related psychopathology.

Reviewer: 2

Reviewer Name: Keely Cheslack-Postava

Institution and Country: New York State Psychiatric Institute/Columbia University
USA

Please state any competing interests or state ‘None declared’: None declared.

Please leave your comments for the authors below

This manuscript examines the relationship between “Common Mental Distress”, a latent dimension of general psychopathology, defined using a factor analytic model, with suicidal thoughts and non-suicidal self-injury in 2 cohorts of adolescents and young adults.

This study has potential important implications for prevention of suicide-related behaviors at a population level. The strengths include a very thorough analysis and documentation, replication (of part of the study) in 2 separate populations, an extensive list of questionnaire items contributing the definition of CMD, and mediation analysis with longitudinal data measured over 3 timepoints.

I have a few main concerns and some additional comments.

1. Models (i.e. association of CMD with ST and NSSI; mediation models) were adjusted for only a very limited set of covariates (i.e. age and sex)... what about SES, family structure and environment, history of adverse events or trauma, to name only a few??

We acknowledge the limited number of controlled confounders in the Limitations (page 12):

“Although ethnicity and socioeconomic status (indicated by IMD) were unrelated to ST and NSSI (Supplementary Tables 3 and 4), and thus were not included in our analyses, we did not control for the effect of other possible confounders such as adverse life experiences, early trauma, family structure or more detailed information about family socio-economic situation (unemployment, poverty etc.).

Of note, when examining mediation, not only confounding of the exposure-outcome association is of concern, but of the exposure-mediator and mediator-outcome associations. In the mediation model, couldn't the baseline (i.e. T1) CMD confound the described associations? It seems a strength of this longitudinal data would be the ability to examine the effects of changes over time in CMD levels (i.e. increase from T1 to T2).

We have not amended the manuscript but respond here:

We have computed a cross-lagged panel model to account for the effects of the CMD at time 1 and time 3 as well as NSSI and ST at time 2, as suggested by the comment above. The results (standardised pathway coefficients with confidence intervals in square brackets) are displayed in the figure below. Indirect effects are reported underneath the figure. The mediation pathways tested in models reported in Figure 3 in the main manuscript and Supplementary figure 2 are here depicted in green colour.

In this cross-lagged panel model, pathways from $NSSI_{T1}$ and ST_{T1} to CMD_{T2} were not significant. Neither were there significant pathways from CMD_{T2} to $NSSI_{T3}$ and ST_{T3} . Significant cross-lagged pathways are depicted in red – these were pathways from CMD_{T1} to $NSSI_{T2}$ and ST_{T2} . We interpret the differences in the models – “pure” mediation model and cross-lagged model below – as an indication that there is very little fluctuation in the levels of CMD over the 3 year period, so this cross-lagged model depicts effects occurring due to error variance and confinement of within- and between-individuals variance, rather than the meaningful associations between variables over time – the problem that has been described in the literature criticising the use of cross-lagged panel models (e.g., Berry et al., 2017; Hamaker et al. 2015; Wu et al., 2018). We therefore suggest that mediation models reported in Figure 3 in the main manuscript and the Supplementary figure 2 are methodologically more accurate than the cross-lagged analysis suggested in the above comment.

References:

Berry, D., Willoughby, M. T. On the Practical Interpretability of Cross-Lagged Panel Models: Rethinking a Developmental Workhorse. *Child Dev.* 2017; 88(4):1186-1206. doi: 10.1111/cdev.12660. Epub 2016 Nov 23.

Hamaker, E. L., Kuiper, R. M., Grasman, R. P. A critique of the cross-lagged panel model. *Psychol Methods.* 2015; 20(1): 102-116.

Wu W., Carroll, I. A., Chen, P. Y. A single-level random-effects cross-lagged panel model for longitudinal mediation analysis. *Behav Res Methods.* 2018;50(5): 2111-2124.

Figure depicting cross-lagged panel model

Indirect effects in cross-lagged model (standardised coefficients and 95% confidence intervals):

- Effect from NSSI_{T1} to NSSI_{T3} via CMD_{T2}: -0.001 [-.011 -- .010]
- Effects from ST_{T1} to NSSI_{T3} via CMD_{T2}: 0.011 [-.012 -- .033]
- Effects from NSSI_{T1} to NSSI_{T3} via NSSI_{T2}: 0.125 [.032 -- .218]
- Effects from ST_{T1} to NSSI_{T3} via NSSI_{T2}: 0.065 [-.015 -- .146]
- Effects from NSSI_{T1} to NSSI_{T3} via ST_{T2}: -0.010 [-.066 -- .045]
- Effects from ST_{T1} to NSSI_{T3} via ST_{T2}: -0.011 [-.117 -- .095]
- Effects from CMD_{T1} to NSSI_{T3} via ST_{T2}: -0.057 [-.308 -- .194]
- Effects from CMD_{T1} to NSSI_{T3} via CMD_{T2}: 0.121 [-.106 -- .348]
- Effects from CMD_{T1} to NSSI_{T3} via NSSI_{T2}: **0.139* [.031 -- .247]**
- Effects from NSSI_{T1} to ST_{T3} via CMD_{T2}: 0.001 [-.015 -- .018]
- Effects from ST_{T1} to ST_{T3} via CMD_{T2}: 0.020 [-.007 -- .046]
- Effects from NSSI_{T1} to ST_{T3} via NSSI_{T2}: 0.062 [-.056 -- .181]
- Effects from ST_{T1} to ST_{T3} via NSSI_{T2}: 0.039 [-.034 -- .113]
- Effects from NSSI_{T1} to ST_{T3} via ST_{T2}: -0.003 [-.066 -- .061]
- Effects from ST_{T1} to ST_{T3} via ST_{T2}: 0.012 [-.108 -- .132]
- Effects from CMD_{T1} to ST_{T3} via ST_{T2}: -0.013 [-.287 -- .261]
- Effects from CMD_{T1} to ST_{T3} via CMD_{T2}: **0.250** [.041 -- .459]**
- Effects from CMD_{T1} to ST_{T3} via NSSI_{T2}: 0.071 [-.061 -- .203]
- Effects from NSSI_{T1} to CMD_{T3} via CMD_{T2}: 0.009 [-.036 -- .054]
- Effects from ST_{T1} to CMD_{T3} via CMD_{T2}: 0.050 [.001 -- .098]
- Effects from NSSI_{T1} to CMD_{T3} via NSSI_{T2}: 0.013 [-.071 -- .098]
- Effects from ST_{T1} to CMD_{T3} via NSSI_{T2}: 0.013 [-.029 -- .056]
- Effects from NSSI_{T1} to CMD_{T3} via ST_{T2}: -0.028 [-.071 -- .016]
- Effects from ST_{T1} to CMD_{T3} via ST_{T2}: -0.055 [-.115 -- .005]
- Effects from CMD_{T1} to CMD_{T3} via ST_{T2}: -0.148 [-.319 -- .023]
- Effects from CMD_{T1} to CMD_{T3} via CMD_{T2}: **0.674** [.568 -- .780]**
- Effects from CMD_{T1} to CMD_{T3} via NSSI_{T2}: 0.016 [-.077 -- .109]

2. P. 12, the first paragraph under “Implications and Conclusions” could use overall re-writing for clarity, in particular “psychopathology is generated in a probabilistic manner...” (generated how?); “this begins to yield clusters ...” (of what?); explain briefly “hierarchical approaches”.

We have amended the text:

We speculate that psychopathological items accumulate in a probabilistic manner rather than in diagnostic clusters, with common phenomena concerning depression and anxiety much more likely to occur before rarer phenomena such as NSSI, ST or psychotic experiences.

3. P. 14, last paragraph. This paragraph seems to delve into detail beyond what is supported by this study and a briefer treatment would seem appropriate (i.e. the evidence presented here suggests the conceptual impact of interventions to shift population distribution of CMD, but does not apply to anything about specific potential interventions, i.e. by digital platforms or social media).

The last paragraph has been shortened (page 14):

Defining putative universal interventions to shift the population distribution of CMD will require careful research that can draw from other areas of medicine such as cardiovascular disease and stroke³⁰. Elements have been widely scoped in the USA¹⁵ and elsewhere, but not for constructs of population health and wellbeing such as CMD. Interventions may involve decreasing common triggers¹⁵ or improving young people's abilities to cope with stressors^{47, 48}.

Note, this comment was in opposition to Reviewer 3 who wanted more extensive discussion. We have adopted the approach suggested by Reviewer 2 to tighten the focus to the limits of the data we present.

4. Many items go into defining the measure of CMD, how does that affect its potential utility?

We acknowledge that the range of items used for assessment of CMD in the current study do not constitute a "ready to use" questionnaire. However, from the conceptual point of view, if the concept has theoretical validity and utility, it can be measured by a broad range of items. There are various psychometric methods of shortening measures by selecting the items with highest contribution to the measured latent concept, e.g., the IRT method (allowing the selection of items with the highest information), or methods of "personalising" psychometric assessment, such as computer-adaptive testing (CAT), allowing the shortening of the time necessary for the psychometric assessment. Further studies could focus on developing a "user-friendly" CMD assessment tool and its psychometric validation.

Minor

1. Abstract, line 26 states "Volunteers age 14-24 years" but the intro section (p. 5) says 14-26 years.

The line "Volunteers age 14-24 years recruited from primary health care..." (page 2, Abstract) refers to the age of participants at time 1, when the study commenced. As the study lasted over 2 years, the final available age poll of participants was 14-26, which the sentence "...the presence of a CMD dimension in young people aged 14-26 years..." (page 5) refers to.

2. Abstract, abbreviations ST and NSSI should be spelled out.

Done (page 2)

3. Introduction, p. 5, lines 33-47. The statement of the aims comes across a bit disjointedly – first, "... we aimed to test here associations between CMD and suicide risk," And then a numbered list of 2 additional questions. It would be easier to follow both here and in the results section if all of the aims could be presented as one cohesive list (either numbered or narratively).

We have amended the format so that our work is now presented as three steps, rather than a preliminary phase followed by analysis to answer two questions.

4. P. 7, line 40, spell out "WLMSV" on first use.

Done (page 7)

Reviewer: 3

Reviewer Name: Karen H. Larwin

Institution and Country: Youngstown State University

Please state any competing interests or state 'None declared': None declared

Please leave your comments for the authors below

Thank you for the opportunity to read and review this important manuscript about Common Mental Distress as a potential indicator of suicide. I have a few considerations for revision.

First, please indicate the full term prior to using the acronym. For example with "NSSI" and "ST" in the abstract. The reader might not know what you are referencing.

Done (page 2).

Second, I believe that in addition to the recommendation to provide interventions to help lower the CMD experienced by these individuals, I would consider adding some discussion about why the CMD might be where it is at and how these individuals might benefit from some greater understanding of lifes ups and downs, reflections on values, or work on mindfulness.

As noted above, this interesting suggestion is in opposition to the approach suggested by Reviewer 2 who wanted us to restrict the discussion; we have done that but consider the referencing covers the ground referred to by Reviewer 3. Shortening the final parts of the discussion emphasises that more research is required to address the important points she raises.

Also, I would like you to comment on whether order-effect was a limitation with the items used. Lastly, did you conduct reliability estimates on any of the data? This information should be provided so that we can understand how well these items worked across the two groups. These are all minor revisions and should be easily managed.

Similar to our answer to comment 4 of Reviewer 2, we have to re-iterate that it was not the aim of the present study to develop and test a psychometric measure of the CMD. Further studies focusing specifically on psychometric validation and calibration of an instrument measuring CMD are needed. Such studies could address the issue of items functioning (including item order-effects or uniform and non-uniform DIFs) and items selection to obtain succinct and precise measurement tool.

VERSION 2 – REVIEW

REVIEWER	Keely Cheslack-Postava New York State Psychiatric Institute/Columbia University
REVIEW RETURNED	21-Nov-2019

GENERAL COMMENTS	The authors have provided thorough responses and adequately addressed my concerns.
--

REVIEWER	Karen H Larwin Youngstown State University, Youngstown, Ohio United State of America
REVIEW RETURNED	04-Dec-2019

GENERAL COMMENTS	This is an important piece of research that will add substantially to the existing discourse. My only request is that you offer more and easier explanation to the reader throughout your results section because I think many from counseling do not have a statistical background. I always ask my students to write their results as
---

	though they were writing for a freshman college student. This change will make this paper more accessible to many readers.
--	--

VERSION 2 – AUTHOR RESPONSE

We have made some changes in the Results section and provided two footnotes to improve the accessibility of the study for clinicians as suggested by Reviewer 3. We would like to point out that we included a longer description of the statistical analysis in the online supplement (mainly due to the limited space in the manuscript). We believe that the Discussion section includes a succinct and clear summary of the results which can be understood by a reader without a statistical background.